# Molecular programs of fibrotic change in aging human lung

Seoyeon Lee [1,8], Mohammad Naimul Islam [2,8], Kaveh Boostanpour[1], Dvir Aran [3], Guangchun Jin[2], Stephanie Christenson[1], Michael A. Matthay [1], Walter L. Eckalbar[1], Daryle J. DePianto[4], Joseph R. Arron [4], Liam Magee[1], Sunita Bhattacharya[2,5], Rei Matsumoto[6], Masaru Kubota[6], Donna L. Farber [6,7], Jahar Bhattacharya [2,8 ✉], Paul J. Wolters [1,8 ✉] & Mallar Bhattacharya [1,8 ✉]

Lung fibrosis is increasingly detected with aging and has been associated with poor outcomes in acute lung injury or infection. However, the molecular programs driving this pro-fibrotic evolution are unclear. Here we profile distal lung samples from healthy human donors across the lifespan. Gene expression profiling by bulk RNAseq reveals both increasing cellular senescence and pro-fibrotic pathway activation with age. Quantitation of telomere length shows progressive shortening with age, which is associated with DNA damage foci and cellular senescence. Cell type deconvolution analysis of the RNAseq data indicates a progressive loss of lung epithelial cells and an increasing proportion of fibroblasts with age. Consistent with this pro-fibrotic profile, second harmonic imaging of aged lungs demonstrates increased density of interstitial collagen as well as decreased alveolar expansion and surfactant secretion. In this work, we reveal the transcriptional and structural features of fibrosis and associated functional impairment in normal lung aging.

[1] Department of Medicine, Division of Pulmonary, Critical Care, Allergy, and Sleep, University of California, San Francisco, CA, USA. [2] Lung Biology Laboratory, Department of Medicine, Division of Pulmonary, Allergy, and Critical Care Medicine, Vagelos College of Physicians and Surgeons of Columbia University, New York, NY, USA. [3] Lorry I. Lokey Interdisciplinary Center for Life Sciences & Engineering, Technion Israel Institute of Technology, Haifa, Israel. [4] Genentech Research and Early Development, Genentech, Inc., South San Francisco, CA, USA. [5] Department of Pediatrics, Vagelos College of Physicians and Surgeons of Columbia University, New York, NY, USA. [6] Department of Surgery, Vagelos College of Physicians and Surgeons of Columbia University, New York, NY, USA. [7] Department of Microbiology and Immunology, Columbia University, New York, NY, USA. [8]These authors contributed equally: Seoyeon Lee, Mohammad Naimul Islam; Jahar Bhattacharya, Paul J. Wolters, and Mallar Bhattacharya. ✉email: jb39@columbia.edu; paul.wolters@ucsf.edu; mallar.bhattacharya@ucsf.edu

Lung capacity and resilience decline and susceptibility to disease increase with age[1], and molecular mechanisms of aging have been implicated in the pathobiology of acute and chronic lung diseases, most of which increase in incidence with age[2]. Thus, targeting respiratory aging therapeutically or prophylactically will require an understanding of lung-specific molecular programs of aging[3–9]. A number of transcriptomic studies of lung aging have been performed, including murine studies[10–12] and one transcriptomic dataset comparing lungs from middle-aged and elderly humans[13]. However, transcriptomic profiling across the adult range, with accompanying structural and functional analyses, is not available. In this work, to characterize the effect of age on the molecular natural history of the lung, we collect human deceased donor lungs not used for clinical transplant for multimodal analysis including gene expression profiling, quantitation of telomere length, and imaging of lung collagen as well as alveolar dynamics. Transcriptomic analysis reveals an increase in senescence and pro-fibrotic pathways as well as cellular evolution, including an increase in the proportion of fibroblasts and a decrease in epithelial cells. Imaging confirms a pro-fibrotic profile with increased interstitial collagen as well as associated impairment in alveolar inflation and surfactant secretion. Taken together, these findings provide a molecular blueprint of pro-fibrotic evolution of the human lung with aging.

## Results

**Bulk RNA-seq of human lung.** Over a 6-year period, we prospectively collected 86 human deceased donor lungs as part of the Lung Aging Cohort (LAC). Lungs were evenly distributed in age between 16 and 76 years (Fig. 1a, b). Donors were not known to have any underlying pulmonary conditions, and gender, smoking status, and ethnicity are summarized in Fig. 1c and detailed in Supplementary Table 1. Tissue samples were harvested from distal lung and frozen in liquid nitrogen on receipt. RNA was later extracted from these samples for Illumina-based sequencing in a single run after cDNA library preparation. The number of uniquely mapped reads ranged from 19.1 million to 42.7 million with an average of 25.3 million reads per sample.

Differential gene expression analysis using a generalized linear model controlling for gender and smoking status identified 22 genes that correlated significantly with age as a continuous variable in the LAC (Fig. 1d). To validate this lung aging gene signature, we used publicly available data from the genotype-tissue expression (GTEx) project of multiple tissues from over 300 individuals[14]. The LAC gene signature was also associated with age in the GTEx lung samples. Interestingly, an overlap of the lung aging gene signature was found for aging of sun-exposed regions of skin, but not for non-sun-exposed skin, kidney, liver, or heart (Fig. S1a).

Given the relevance of cellular senescence to aging, samples were interrogated for evidence of cell senescence markers and pathways. The canonical senescence marker p16 (CDKN2A) was among the most highly upregulated genes in aging lungs (Fig. 2a). To further assess for senescence reprogramming, we asked whether a consensus senescence gene signature that we recently defined by RNA-seq of senescent lung epithelial cells and fibroblasts[15] was upregulated in the LAC. The consensus senescence gene signature was increased with age in both the LAC and GTEx Lung datasets (Fig. 2b). We then performed ingenuity pathway analysis (IPA) of genes associated with aging in the LAC (Fig. 2c and Supplementary Data 1), identifying pathways that were largely validated in GTEx lung (Fig. S1b). Consistent with cellular senescence, cell proliferation pathways were inhibited, both by IPA and by Gene Ontology analysis[16,17],

and both upstream regulators p16 (CDKN2A) and p21 (CDKN1A) were activated in aged lungs (Fig. 2c; Fig. S1c; and Supplementary Table 2). Interestingly, cell death pathways were also prominently activated. To ensure that the effect of time between cross-clamp and cryopreservation was not a confounding factor, first, we confirmed that there is no correlation between age and time to cryopreservation for samples with available data (Fig. S1d). We have also previously reported in a study of 99 lungs that cold ischemia time did not have any effect on the level of cytokines produced by the intact lung in response to pro-inflammatory stimuli delivered by whole-lung perfusion[18]. Next, correlation analyses were done to confirm that time to cryopreservation did not correlate with gene expression of the senescence markers or the apoptosis and necrosis signatures from IPA (Fig. S1d). In addition, we performed gamma-H2AX immunohistochemistry on two subsamples with divergent times to cryopreservation and found no correlation between the time to cryopreservation and the level of DNA damage (Fig. S1e). Overall, the marker-based analysis and IPA suggest activation of senescence and cell stress in aged lungs at steady state.

Cellular senescence has been associated with fibrotic lung disease, in part due to a senescence-associated secretory profile that has pro-fibrotic effects[19–25]. Therefore, we considered that cell senescence may also underlie aging-associated subclinical interstitial fibrosis, or interstitial lung abnormalities, a phenomenon recognized recently by radiographic studies of asymptomatic aged individuals[26–28]. These radiographic findings of fibrosis have been correlated with histopathologic features of fibrosis, including fibroblastic foci and subpleural distribution[29]. However, little is known about the molecular and cellular programs responsible for the pro-fibrotic evolution in the aging lungs. We first noted that by IPA, pathways consistent with mesenchymal activation and fibrosis (TGF-beta pathway mediators and the epithelial-to-mesenchymal transition regulator TWIST1) were activated in aged lungs (Fig. 2c and Supplementary Data 1). These results were largely confirmed in the GTEx lung dataset but not consistently in other organs (Fig. S1b). Furthermore, several of the most highly upregulated genes with age in the LAC (Fig. 1d) have known pro-fibrotic effects; for example, RSPO4 has been associated with a decline in lung function in patients with lung fibrosis[30]. We also performed pathway activation level analysis utilizing over 3044 human molecular pathways extracted from the Biocarta, Reactome, KEGG, Qiagen Pathway Central, NCI, and HumanCYC databases[31], comparing the oldest and youngest quintiles in the LAC and GTEx lung, and found that multiple growth factor pathways implicated in fibrosis, such as PDGF, FGF, LPA, and ephrin A[32–35], were upregulated in the aged lungs (Fig. S1g and Supplementary Data 2).

**Quantitation of telomere length.** Next, since a major cell-intrinsic driver of senescence is telomere shortening, we isolated genomic DNA from the lung samples and used a quantitative PCR-based assay to measure telomere length. This analysis revealed that average lung telomere length progressively decreased across the lifespan (Fig. 2d). Telomere attrition leads to telomere uncapping, which triggers a DNA damage response including p53 activation[36]. To test the significance of telomere shortening to cellular states, gene expression was compared between subsamples that were significantly different in telomere length but approximately matched by age. IPA upstream regulator analysis of differentially expressed genes revealed that the canonical senescence regulators p53 and p16 were activated in association with decreased telomere length; furthermore, sites of DNA damage were significantly increased across multiple high

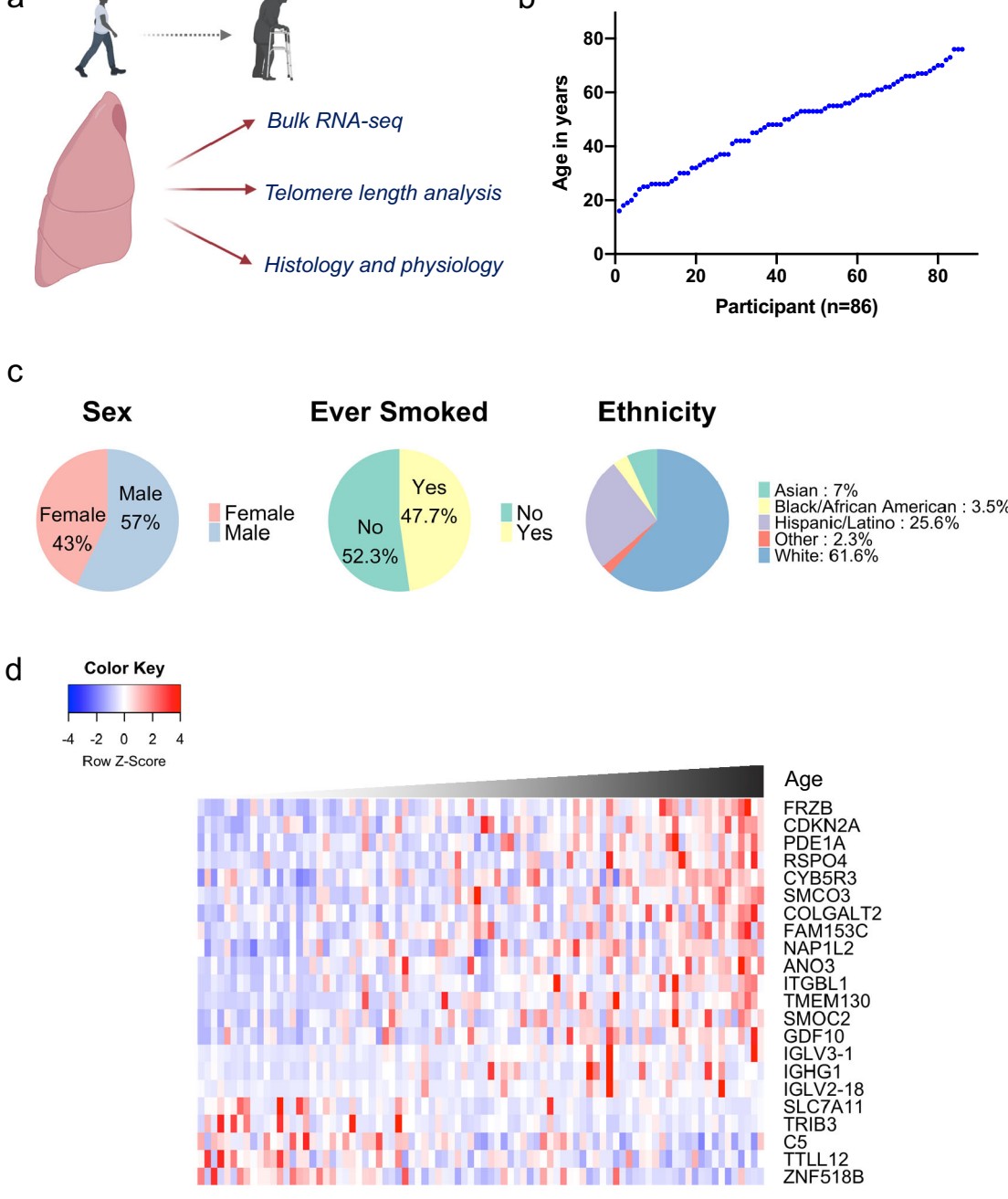

**Fig. 1 RNA-seq reveals a lung-specific aging signature. a** Schematic of the lung aging cohort (LAC), a study of human lungs varying in age across the adult lifespan and profiled with multiple approaches. **b** Participant age plotted against increasing order of age. **c** Demographic features. **d** Heatmap of gene expression by bulk RNA-seq of the distal lung. Genes listed were significantly correlated with age in a generalized linear model adjusted for smoking and gender (FDR $p \leq 0.1$ by DESeq2 two-sided likelihood ratio test). Z-scores represent within-gene relative expression across samples.

power fields for the short- compared to long-telomere samples by gamma-H2ax immunohistochemistry (Fig. 2d), with a similar trend when the data for individuals were averaged (Fig. S1f). These results suggest that age-associated lung telomere attrition likely contributes to the senescence profile observed across the sample.

**Cellular composition of the aging lung.** Given the senescence and cell death profiles revealed by our analysis, we next asked whether lung aging is associated with changes in the cellular composition of the lung. To address this question, cell type deconvolution analysis was performed on the bulk RNA-seq data.

First, SingleR[37] was used to annotate cell types from published single-cell transcriptomes for three young, healthy human lungs[38]. Differential gene expression analysis confirmed characteristic markers for lung epithelial cells and fibroblasts (Supplementary Data 3 and 4, respectively). MuSiC[39] was then applied to these SingleR-identified clusters of cell subtypes to deconvolve proportions of cell types in each LAC and GTEx lung sample. Interestingly, the proportion of epithelial cells declined with age; on the other hand, the proportion of fibroblasts increased, consistent with a fibrotic change in the aging lung (Fig. 3a and Fig. S2a). Given the similarities in gene expression and pathways between the lung and sun-exposed skin, we also performed cell type deconvolution analysis on the GTEx skin datasets with cell

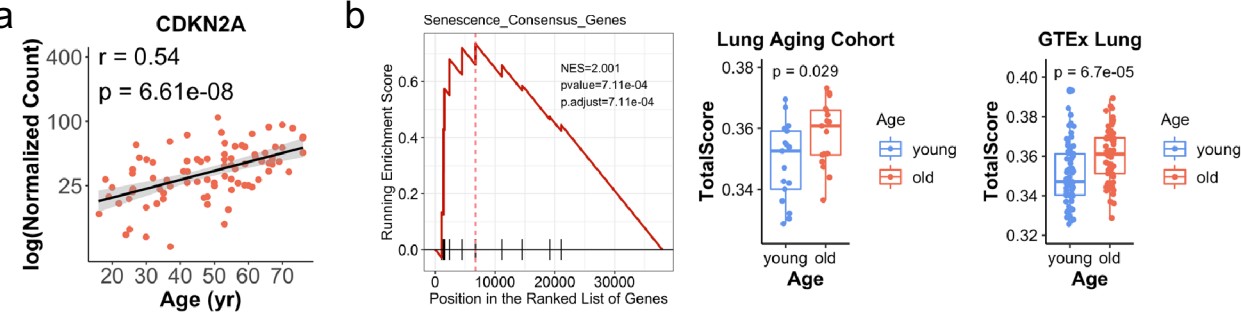

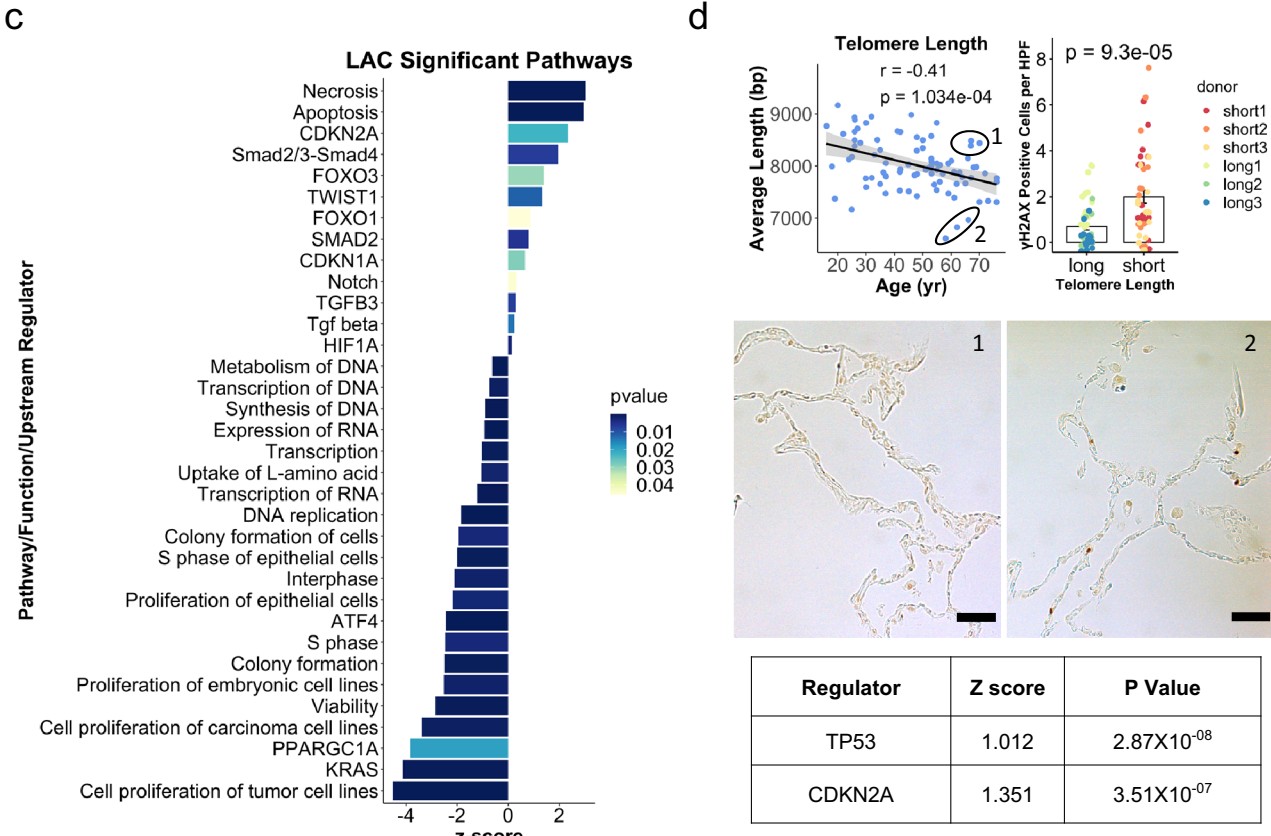

**Fig. 2 Lung cellular senescence increases with aging. a** p16 expression by age in the LAC (log scale). Pearson *R* is shown. The gray band represents the 95% confidence interval. **b** Gene set enrichment analysis of 11 consensus senescence markers from DePianto et al[15]. Genes were ranked by Pearson *R* between normalized gene expression and age. Normalized enrichment score, *P* value, and adjusted *P* value are shown. Single sample gene set enrichment analysis using the 11-gene signature is shown for individuals in the LAC (*N* = 17 biologically independent samples in each group) and GTEx lung (*N* = 69 biologically independent samples in each group). *P* values are for the two-sided Student's *t*-test. For the boxplots, the middle line shows the median, the lower and upper hinges are the first and third quartiles, and the lower and upper whiskers extend to the value at most 1.5 IQR below the first quartile or 1.5 IQR above the third quartile, respectively. Data beyond the end of the whiskers are outlying points. **c** Ingenuity pathway analysis (IPA) computed with the genes from LAC that were correlated with age at *p* ≤ 0.05 level of significance. *P* values are for right-tailed Fisher's exact test without adjustment for multiple comparisons. **d** qPCR-quantified telomere length plotted by age for the LAC (*N* = 84 biologically independent samples). Pearson *R* is shown. The gray band represents the 95% confidence interval. The circled subsamples differ in telomere length (1 = long, 2 = short) and were used for γH2AX immunohistochemistry, with representative images below (Scale bars: 25 μm) and quantification to the right (*N* = 3 samples in each group, *N* = 15 images examined per sample). Each point represents the number of γH2AX+ cells within a separate image. *P* value is for a two-sided Welch's *t*-test comparing all values for the long telomere group versus the short. Data were presented as mean values ± SEM. Differential gene expression of the subsamples by bulk RNA-seq was used for IPA. The predicted activity of upstream senescence regulators in the short telomere subgroup, with *Z*-scores and *P* values, is shown in the table.

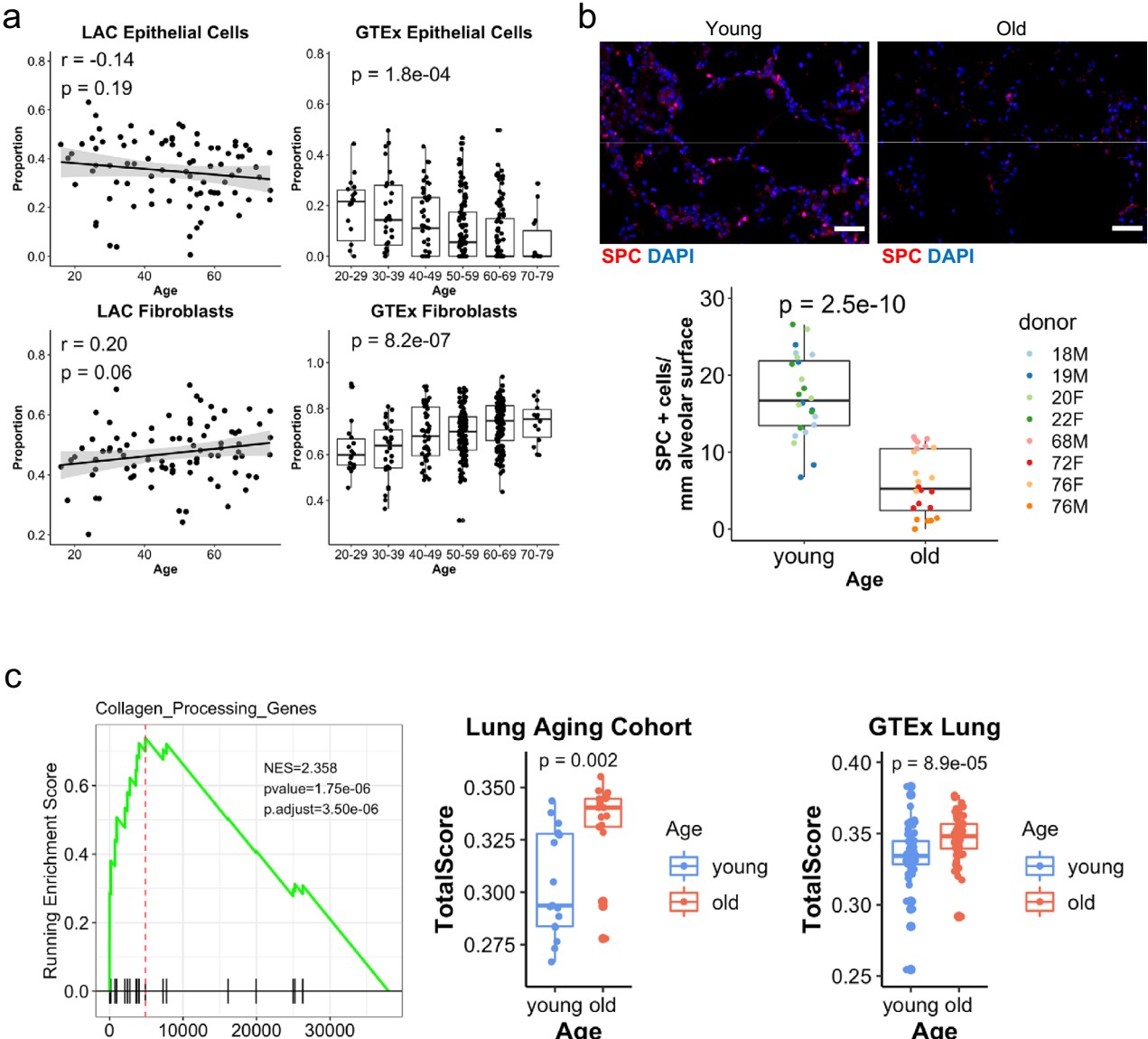

**Fig. 3 Fibrotic programs in lung aging. a** Cell type deconvolution of bulk RNA-seq data from the LAC ($N = 86$ biologically independent samples) and GTEx lung ($N = 345$ biologically independent samples). Age for GTEx is represented in decades. Pearson $R$ and $P$ values are shown for LAC, and one-way ANOVA is shown for GTEx. The gray bands represent 95% confidence intervals. **b** Quantification of type 2 cells in old and young lungs by labeling of SPC + cells, with representative images (Scale bars: 50 μm) ($N = 4$ biologically independent samples in each group, $N = 6$ images examined per sample). Donor age and gender are shown. Each point represents the number of SPC + cells in a separate image, adjusted for alveolar membrane length. $P$ value is for a two-tailed Student's $t$-test comparing all dots from the old versus the young. **c** Collagen-processing gene expression in the LAC by gene set enrichment analysis. Genes were ranked by Pearson $R$ between normalized gene expression and age. Normalized enrichment score, $P$ value, and adjusted $P$ value are shown. Single sample gene set enrichment analysis scores are plotted for the genes enriched in aging (genes listed in Fig. S3a) in the oldest and youngest quintiles from the LAC ($N = 17$ biologically independent samples in each group) and GTEx lung ($N = 69$ biologically independent samples in each group). $P$ values are for the two-tailed Student's $t$-test. For boxplots in **a–c**, the middle line shows the median, the lower and upper hinges are the first and third quartiles, and the lower and upper whiskers extend to the value at most 1.5 IQR below the first quartile or 1.5 IQR above the third quartile, respectively. Data beyond the end of the whiskers are outlying points.

types defined with SingleR[37] for a skin single-cell RNA-seq dataset[40]. Interestingly, there was a similar increase of fibroblasts and a decrease of epithelial cells in sun-exposed skin but not in non-sun-exposed skin (Fig. S2b). Within the lung epithelial compartment, we found specifically alveolar type 2 cells to decrease with age by applying MuSiC with a single-cell dataset[41] that was pre-annotated with more granular cell subtypes (Fig. S2c). The decrease in alveolar type 2 cells was also confirmed by immunostaining for the type 2 cell marker pro-SPC (Fig. 3b). Type 2 cells are thought to be necessary for epithelial renewal in

the lung, even at steady state[42]. Collectively, these findings support a DNA damage response resulting from telomere shortening, leading to epithelial senescence and pro-fibrotic pathway activation characterized by expansion of the mesenchyme in the aging human lung.

**Collagen-processing genes in lung aging.** We noted no robust differences in the expression of collagen genes between old and young. However, collagen accumulation is due not simply to excess collagen deposition, but also to an imbalance of collagen

production and destruction, as well as changes in extracellular collagen structure and stability[43,44]. Therefore, the oldest and youngest quintiles in the LAC were examined for expression of genes known to regulate posttranslational processing of collagen, including lysyl oxidases, transglutaminases, and tissue inhibitors of matrix metalloproteases, which inhibit collagen turnover by metalloproteases. Remarkably, a large proportion of these genes were upregulated with age (Fig. 3c and Fig. S3a). We then tested whether the age-associated subset of these genes from the LAC could be validated in other datasets and found robust upregulation of the signature in the GTEx lung cohort (Fig. 3c), and for GTEx sun-exposed skin, but not for multiple other organs (Fig. S3b). To test whether the collagen-regulatory gene signature (Fig. S3a) or the lung aging-associated gene set (Fig. 1d) could be therapeutically targeted, we used the Connectivity Map dataset, a library of gene expression profiles induced in multiple cell types by small molecules. Our analysis revealed several candidate compounds with signature-reversing properties (Fig. S3c and Supplementary Data 5).

**Live lung imaging of collagen and alveolar dynamics**. These changes in gene expression and cellular content of the lung led us to test the age dependence of collagen structure and distribution in young and aged human donor lungs by live two-photon microscopy. Second harmonic (SH) generation by extracellular collagen has been used to visualize the fibrillary structure of collagen in fixed tissues[45]. Here we applied the technique to live, unfixed human lungs. SH imaging revealed marked differences in the collagen pattern in young versus aged lungs in the subpleural space. In SH images of young lungs (age <40 years), well-defined collagen fibers of 1-micron thickness were regularly evident with interfibrillar spaces of 3–5 microns (Fig. 4ai). Fluorescence analyses along lines drawn on the x-y planes of these images revealed a fluorescent spike where the analysis line intersected a fibril, while the low inter-spike fluorescence reflected the collagen-free interfibrillar space (Fig. 4b). In aged lungs (age >50 years) a similar fibrillar pattern was also evident in some regions, but in other regions the spiked fibrillar pattern was notably absent (Fig. 4aii, b). In these regions line analyses revealed dense packing of considerably thinner fibers (Fig. 4b). Area analysis of SH fluorescence quantified in the Z direction starting at the pleural surface revealed Gaussian distributions of fluorescence intensity reflecting intensity and depth of subpleural collagen deposition (Fig. 4c). Notably, subpleural collagen density varied considerably between different regions of the same lung for both young and old donors, as indicated by the spread of density values for each lung (Fig. 4d). On average collagen density was higher in the older age group (Fig. 4e). In the alveolar interstitium subjacent to the pleural space, collagen density was about ten times less than in the subpleural region (Fig. 4f). However, here too older lungs had higher interstitial collagen density (Fig. 4g).

To determine whether increased collagen density impeded alveolar expansion, we carried out optical quantification of alveolar dimensions at low and high transpulmonary pressures (Fig. 4h). This analysis demonstrated that there was a monotonic loss of alveolar expansion with age (Fig. 4i) and that this loss correlated with collagen density (Fig. 4j). Thus with age, alveolar expansion was limited by a constraining effect of increased perialveolar and subpleural collagen. The alveolar expansion causes secretion of surfactant, which maintains alveolar patency and provides epithelial defense against inhaled pathogens. To quantify the variability of surfactant secretion, a fluorescence approach was used to quantify surfactant secretion at the single alveolar level[46]. Our data indicate a strong negative correlation of surfactant secretion with age and with collagen density (Fig. 4k–m).

## Discussion

Our study results provide evidence of pro-fibrotic transcriptomic and structural change in human lung aging. These findings can be viewed from the perspective of the respiratory vulnerability associated with aging. For example, acute respiratory distress syndrome (ARDS), a major cause of mortality from acute lung infections and injury, is associated with worse outcomes in individuals with the subclinical lung fibrosis seen with aging;[47] of note, telomere shortening in peripheral blood leukocytes has also been associated with increased severity and mortality from ARDS[48]. Furthermore, our findings implicate a mechanism in common with pathologic lung fibrosis, where aging and associated telomere shortening are thought to be a root cause[20,22]. In our study, the decreasing lung telomere length with age taken together with upregulation of senescence markers with shorter telomeres provides evidence for the role of telomere shortening in driving the senescence profile of normal human lung aging. These data suggest a link between normal lung aging and pathologic fibrosis along the spectrum of telomere length-regulatory mechanisms. Moreover, cellular senescence has also been implicated in pathologic lung fibrosis, and we confirm the upregulation of both canonical and novel markers of senescence[15] in lung aging. We also note that senescence has been recognized as an important feature of chronic obstructive pulmonary disease (COPD), where telomere dysfunction or shortening has been associated with DNA damage foci and expression of senescence markers[49,50]. Under which conditions senescence in the aging lung may contribute to the severity of lung fibrosis or COPD or both, could be addressed by prospective studies comparing healthy, fibrotic, and COPD cohorts directly.

Our data address an unmet need in the field of cellular senescence of advancing the study of senescence from cultured cells and animal models to human tissues[40]. The increase in senescence may explain the changes in cellular composition observed, namely decreased alveolar type 2 cells and increased fibroblasts, although future studies using single-cell methods could map the distribution of senescence across cell types. Nonetheless, the increase in fibroblasts together with increased expression of collagen-regulatory genes such as Tgm-2[51] and Loxl1[52] found to have pro-fibrotic effects in mouse models reveals a pattern of fibrotic evolution in normal lung aging. This pro-fibrotic molecular imprint corresponded to fibrotic changes by second harmonic imaging. Alveoli serve as essential gas exchange units that must inflate for effective ventilation, a process that deteriorates with age and can be limited by fibrosis[53]. Furthermore, our live lung imaging revealed that regions of interstitial fibrosis in aged lungs were associated with local alveolar dysfunction, including decreased alveolar expansion and decreased surfactant secretion. The linear structure–function relationship we report reveals a previously unknown inhibitory effect of perialveolar collagen accumulation on surfactant secretion. We were able to establish this understanding because we directly viewed the live lung by optical microscopy and could therefore quantify type 2 cell functionality across a wide range of collagen expression in different alveoli.

There are several limitations to our study. For live optical imaging, we were restricted by the stage dimensions of the microscope to viewing the relatively thin lingular lobe and could not accommodate the bulkier upper and lower lobes under the objective. Despite these constraints, we believe our findings illustrate the use of structural analysis to better understand the lung's aging responses from a functional standpoint. We point out that our SHG findings are alveolus-specific and do not apply to airway collagen, which is unlikely to be a determinant of type 2 cell secretion. Our study does not distinguish chronological aging from the effects of environmental insults accumulated over the

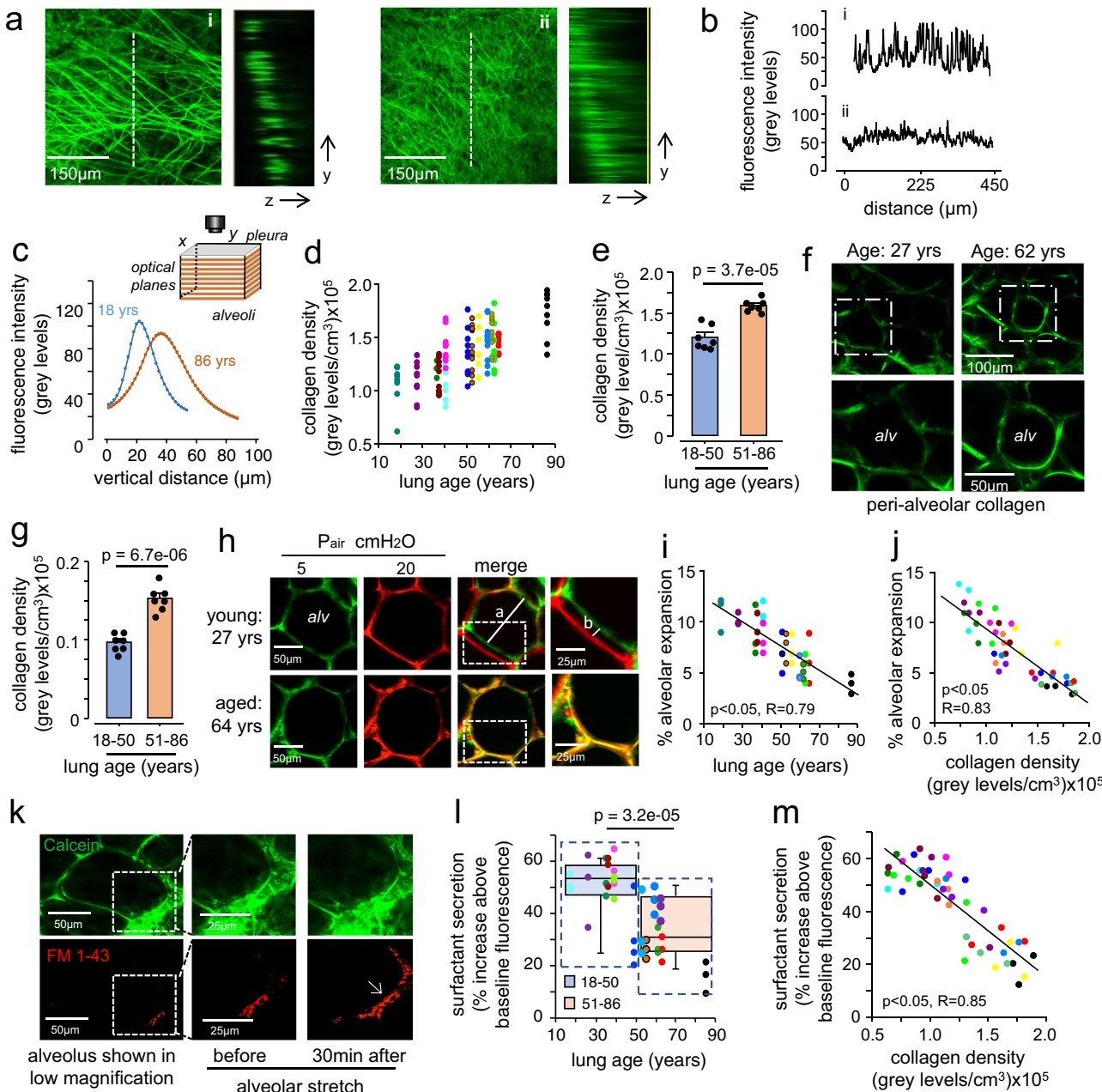

**Fig. 4 Aging-associated fibrosis limits alveolar expansion and surfactant secretion. a, b** Two-photon images (i and ii) show collagen fluorescence by second harmonic generation in the subpleural interstitium of an 18 (left) and an 86 (right) year-old lung. Adjacent panels show collagen fluorescence in the depth plane (y-z) along the indicated lines (dashed lines). Tracings in (**b**) represent fluorescence intensity along the lengths of the dashed lines ("distance"). **c**–**e** Imaging was carried out across a tissue volume calculated as the product of the area and the depth of the imaged field (see sketch). Tracings in (**c**) are from two representative fields and quantify collagen fluorescence along the depth axis from the pleural margin. In (**d**) "collagen density" was calculated as the summed fluorescence per unit volume for multiple fields in each lung. (Each color indicates a separate lung.) Group data are shown in (**e**). **f, g** Images and group data show peri-alveolar collagen. **h** Confocal images show an alveolus (alv) stained with the intracellular dye, calcein-AM. Alveoli were imaged at alveolar pressures 5 (green) and 20 cmH2O (red). The alveolar diameter at 5 cmH2O is marked by the line "a". The septal displacement at 20 cmH2O is marked by the line "b" (see a magnified image of the region selected by the rectangle). **i, j** Alveolar expansion was calculated from the distances of the lines exemplified in **h**, as (b/a)%, and plotted for individual imaged fields against donor age (**i**) and subpleural collagen densities (**j**). **k** The images show an alveolus stained with calcein-AM (green) and the extracellular lipid dye FM1-43 (red). A selected region (rectangle in left image) was magnified in the middle and right images. Alveolar stretch caused surfactant secretion as indicated by a time-dependent increase of red fluorescence (arrow). **l, m** Plots show determinants of surfactant secretion. For **a**: determinations replicated 9 times in an 18- and an 86-year-old lung gave similar results. For **e** and **g**: data were mean ± SEM; n = 7 lungs each group; P value is for two-tailed Student's t-test. For **i, j, l,** and **m**: each dot shows the response for a single alveolus. For **f, h,** and **k**: determinations were repeated in seven lungs in each group. For **i, j,** and **m**: P values were computed by linear regression. For boxplots in **l**: the middle line shows the median, the lower and upper hinges are the first and third quartiles, the lower and upper whiskers extend to the value at most 1.5 IQR below the first quartile or 1.5 IQR above the third quartile, respectively, the data beyond the ends of the whiskers are outlying points, and the P value shown is for two-tailed Student's t-test.

lifespan, which are likely to be relevant given the many genes and pathways found to be in common with sun-exposed skin.

Our study sheds light on lung aging, making use of transcriptomic as well functional analyses from three cohorts spanning the adult age range including over 400 individuals. The perspective gained by our findings is that cellular senescence may drive lung dysfunction. That is, our finding of senescence and fibrosis signatures by RNA-seq and physiologic dysfunction associated with increased collagen density by microscopy reveals senescence as a potential vulnerability factor present even in the healthy lung as it ages. Future studies might build on these findings by exploring the functional effects of the identified biomarkers and mediators of lung aging, determining the relative weight of cell-autonomous and environmental effects, and testing whether preexisting cellular senescence is a predictive factor for the development of, or contributes to worse outcomes in, acute and chronic lung diseases.

## Methods

**Participants**. RNA-seq, type 2 cell immunofluorescence, and telomere length analyses were done with the Lung Aging Cohort, which consists of 86 donor lungs collected between 2012 and 2018 and made available by the Donor West Network[54]. The samples are from lungs that were deemed not suitable for clinical transplant by the attending transplant surgeon, which could be for a number of reasons including but not limited to: age; no suitable recipient was available at the time; local areas of contusion or lung injury in lobes not sampled for the present research but invalidating the organ as a whole for clinical transplant. Fresh tissue fragments were snap-frozen in liquid nitrogen upon receipt. Age, sex, ethnicity, smoking status, and cause of death were recorded. Second harmonic microscopy and surfactant studies were done with intact human lungs obtained from brain dead organ donors at the time of tissue acquisition for transplantation and when not used for clinical transplant, as described in refs. [55–57] through collaboration and protocol with LiveOnNY, the organ procurement organization for the New York area. Demographic data are detailed in Supplementary Table 1.

**Bulk RNA sequencing**. Total RNA was isolated using the miRNeasy Mini Kit (Qiagen). Extracted RNA samples were sent to Novogene for library construction and sequencing. Quantitation and quality control were done in three steps including NanoDrop (Thermo Fisher Scientific), agarose gel electrophoresis, and Agilent 2100 Bioanalyzer (Agilent Technologies). mRNA was enriched using oligo(dT) beads using the NEBNext Poly(A) mRNA Magnetic Isolation Module (New England Biolabs), followed by random fragmentation. cDNA was synthesized using the NEBNext Ultra II RNA Library Prep Kit for Illumina (New England Biolabs). Purified and processed cDNA libraries were checked on Agilent 2100 for insert size and quantified on Qubit and by qPCR. PE 150 bp sequencing was done on Novaseq6000 machines. Adapter trimming and alignment to the reference genome were done using STAR software[58]. Multi-factor differential expression analysis for age, smoking, and gender was done with DESeq2 and the likelihood ratio test was used for hypothesis testing[59].

**Telomere length qPCR**. Genomic DNA was isolated from snap-frozen lung tissues using the Gentra Puregene Kit (Qiagen). DNA samples were run on 1% agarose gel electrophoresis for quality control and quantified using the NanoDrop spectrophotometer (Thermo Fisher Scientific Inc.). For each sample, cycle threshold values for telomere and the reference housekeeping gene (36B4) were determined in triplicates using quantitative PCR[60]. Primer sequences and PCR conditions are detailed in Supplementary Table 3. Delta Ct was calculated by subtracting the mean telomere cycle threshold from the mean 36B4 cycle threshold. Three cell line standards with known telomere lengths[61] (T47D, DU4475, and 1806) were used to graph a standard curve, from which sample telomere lengths were calculated. Samples with a standard deviation of triplicates higher than 0.25 were excluded.

**Pathway analysis**. Pathway analysis was done using the ingenuity pathway analysis software (Qiagen). Differentially expressed genes with $p \leq 0.05$ were used for analysis. Analysis results with $p \leq 0.05$ were considered significant. For the PANTHER Gene Ontology analysis[16,17], differentially expressed genes with $p \leq 0.05$ were used for the LAC. For validation of the LAC Gene Ontology analysis, we used differentially expressed genes from the GTEx lung with $p$ adjusted $\leq 0.05$ since the GTEx dataset is much larger ($n = 345$) than the LAC dataset ($n = 86$) and thus better powered to detect differences. Calculation of pathway activation levels was done using the oncoboxlib software[31]. Results with $p \leq 0.05$ were considered significant.

**Gene set enrichment analysis, ssGSEA, and CMAP**. Gene set enrichment analysis was done using the DOSE package[62]. Pearson metric was used for ranking

genes, and a weighted enrichment statistic was used. Single sample gene set enrichment scores were computed on R using Singscore[63]. CMAP[64] drug perturbation analysis was done using the PharmacoGx[65] package on R.

**Analysis of publicly available data**. The genotype-tissue expression (GTEx) Project[14] data (release V8) used for the analyses were obtained from the GTEx Portal [https://GTExportal.org/] on 8/20/2020. RNA-seq gene read counts, sample attributes, and subject phenotypes were downloaded for differential expression and subsequent analyses. Human lung single-cell data used for cell type deconvolution analysis were downloaded from Reyfman et al.[38] (GSE122960 [https://www.ncbi.nlm.nih.gov/geo/query/acc.cgi?acc=GSE122960]) and Travaglini et al.[41] [https://www.synapse.org/#!Synapse:syn21041850]. Skin single-cell data was downloaded from He et al.[40] (GSE147424 [https://www.ncbi.nlm.nih.gov/geo/query/acc.cgi?acc=GSE147424]).

**Immunohistochemistry and immunofluorescence**. Lungs were fixed with 10% formalin overnight and transferred to 70% ethanol before embedding in paraffin and sectioning to 4 μm thickness. For immunohistochemistry of gamma-h2ax, sections were deparaffinized in xylene and rehydrated in an ethanol gradient series. Antigen retrieval was done by microwaving for 6 min in citrate buffer pH 6 (Sigma). After quenching in 3% $H_2O_2$ in methanol, sections were permeabilized in 0.5% Triton X-100 in PBS. Sections were blocked in 3% BSA, 0.1% Triton X-100, 5% normal goat serum in PBS for 1 h, and incubated at 4 °C overnight with primary antibody (Biolegend, cat. 613402, dilution 1:500), then at room temperature for 3 h with secondary antibody (Santa Cruz Biotechnology, cat. sc-2005, 1:1000). Sections were developed in a DAB working solution (Vector Laboratories) for 8 min, washed, dehydrated, and mounted with Cytoseal. All washes between steps were done with 1X PBS.

For immunofluorescence of pro-SPC, sections were deparaffinized, antigen retrieved, and permeabilized. Sections were blocked in 3% BSA, 0.1% Triton X-100, 5% normal donkey serum in PBS for 1 h, and incubated at 4 °C overnight with primary antibody (EMD Millipore, cat. Ab3786, 1:300), then at room temperature for 1 h with Alexa Fluor 594-conjugated secondary antibody (Life Technologies, cat. A21207, 1:1000). Sections were then washed and mounted with a mounting medium with DAPI (Vector Laboratories). All washes between steps were done with 1X PBS on Day 1, and PBST (1:1000) on Day 2. Images were acquired on a Zeiss Axioscope 5 microscope. Immunoreactive cells were counted while blinded to the ages of the immunostained lungs.

**Cell type deconvolution**. Cell type deconvolution of bulk RNA-seq data from the LAC was performed with MuSiC[39], a publicly available computational resource. ScRNA-seq data from three young donor lungs published by Reyfman et al.[38], three donor lungs published by Travaglini et al.[41], and eight healthy skin biopsies published by He et al.[40] were clustered by Seurat[66] followed by the MuSiC workflow for cell type proportion analysis. Data from Reyfman et al.[38] and He et al.[40] were annotated for cell type by SingleR[37], whereas cell type annotations provided by the authors were used for Travaglini et al.[41].

**Live two-photon imaging of human lungs**. Two-photon and confocal microscopy were carried out on live, de-identified human lungs obtained after ~20 h of cold ischemia. The lingular lobe, which provides a flat surface convenient for live microscopy, was positioned below the objective of a two-photon microscope (TCS SP8, Leica). The lobe's pulmonary artery pressure was held at 10 cmH2O, while the lobe was inflated at alveolar pressure of 5 cmH2O through a bronchial cannula. We subpleurally injected fluorescent dyes through a 31-gauge needle. We detected subpleural collagen as the fluorescence of second harmonic generation (SHG) at excitation and emission wavelengths of 830 nm and 425–460 nm, respectively. Nonspecific autofluorescence and photobleaching were eliminated by an appropriate gain setting. We quantified subpleural collagen density as the integrated collagen fluorescence per cubic centimeter in the space between the visceral pleura and the alveolar epithelium. Stretch-induced surfactant secretion was initiated by single 15-s hyperinflation induced by increasing airway pressure from 5 to 15 cmH2O. We quantified alveolar expansion as described in ref. [67] and surfactant secretion by the timed appearance of lipid-sensitive fluorescence in the alveolar space[46].

**Statistical analysis**. Statistical analysis for comparison of two groups was done using the unpaired, two-sided, two-samples $t$-test. Welch's $t$-test was employed when appropriate. For the comparison of multiple groups, one-way ANOVA was used. Pearson correlation coefficient $R$ was calculated to assess the association of two continuous variables. Unless otherwise stated, a $p$ value less than 0.05 was considered significant. Multiple hypothesis testing using Benjamini–Hochberg method was done when appropriate.

**Study approval**. As confirmed by the UCSF and Columbia University IRBs, per the NIH policy, because all samples in the study including both UCSF and Columbia were acquired from deceased individuals, the study is not considered human subjects research. This policy is based on United States Department of

Health and Human Services human subject regulations under 45 CFR 46 wherein a human subject is defined as "a living individual about whom an investigator (whether professional or student) conducting research obtains (1) data through intervention or interaction with the individual or (2) identifiable private information".

**Patient consent**. In every case, patient surrogates (individuals with power of attorney/next of kin) have provided written consent to the use of deceased donor organs in research. Research, including consent for research, was conducted in alignment with the Declaration of Helsinki.

**Privacy protection**. None of the investigators had access to identifiable private information, and all samples were assigned unique, non-identifying IDs on receipt at both Columbia and UCSF. Their correspondence to the United Network for Organ Sharing (UNOS) IDs assigned by the organ procurement organizations (OPOs) Donor Network West (DNW) and LiveOnNY was not communicated to the OPOs by the investigators, and they are maintained by the investigators under a secure, password-protected network.

**Reporting Summary**. Further information on research design is available in the Nature Research Reporting Summary linked to this article.

## Data availability

Source data are provided with this paper. The RNA-seq data generated in this study have been deposited at NCBI GEO under GSE165192 [https://www.ncbi.nlm.nih.gov/geo/query/acc.cgi?acc=GSE165192]. The processed RNA-Seq data are publicly available at GEO. The raw RNA-seq FASTQ data are protected and are not available at GEO due to data privacy laws. They will be accessible upon request from the NCBI Sequence Read Archive dbGaP under accession code phs002484.v1.p1 [http://www.ncbi.nlm.nih.gov/projects/gap/cgi-bin/study.cgi?study_id=phs002484.v1.p1]. Potential users will need to first contact the corresponding authors to enter into a collaboration agreement for submitting a data access request on dbGaP. For the request, the user must provide documentation of local IRB approval, agree to make results of studies using the data available to the larger scientific community, and provide a letter of collaboration with the primary study investigators. Once requested, the NIH processing timeframe is ~14 working days. Human lung and skin single-cell data used for cell type deconvolution analysis were downloaded from GSE122960 [https://www.ncbi.nlm.nih.gov/geo/query/acc.cgi?acc=GSE122960], GSE147424 [https://www.ncbi.nlm.nih.gov/geo/query/acc.cgi?acc=GSE147424], and Synapse [https://www.synapse.org/#!Synapse:syn21041850]. Source data are provided with this paper.

## Code availability

All code used for analyses is deposited at Github [https://github.com/jasminelee63/AgingLungBulkRNASeq][68].

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

## Acknowledgements

This work was supported by NIH HL131560 and institutional startup funds to M.B., NIH HL139897 to P.J.W., HL145547 to D.L.F. and J.B., NIH HL36024 and HL57556 to J.B., and the Nina Ireland Program for Lung Health to M.B. and P.J.W. Figure 1a was created with Biorender.com.

## Author contributions

S.L. did telomere length measurement and immunofluorescence and performed computational analyses with help from K.B. and L.M. and under the supervision of M.B., D.A., W.L.E. and S.C. D.J.D. and J.R.A. derived and helped S.L. in applying the consensus senescence marker analysis. R.M. and M.K. procured LiveOnNY donor network lungs under the supervision of D.L.F. M.N.I. performed second harmonic and surfactant microscopy with the help of G.J. under the supervision of J.B., S.B. and D.L.F. M.B., P.J.W. and J.B. conceived of the work, supervised experimental planning and analysis, and co-wrote the manuscript with input from M.M.

## Competing interests

D.J.D. and J.R.A. are current employees of Genentech and shareholders in Roche. The remaining authors declare no competing interests.
