## [Peer Review File · Nature Communications]

REVIEWER COMMENTS

Reviewer #1 (Remarks to the Author):

Lee et al. investigate lung biopsies from 86 human donors of different age. The authors conduct RNA sequencing from organ biopsies as well as single sequencing analysis. To validate the data, the authors compare their analysis with publicly available data. The authors describe an upregulation of senescence markers, such as cell cycle inhibitor (Cdkn1a, Cdkn2a) and gammaH2aX (a marker of DNA damage), and telomere shortening. These well-known biomarkers of tissue aging correlate with very slight decrease in epithelial cells vs. fibroblast in the lung biopsies during aging. Given the known role of collagen deposition in aging tissues and lung, the authors also investigate collagen-regulating genes although these were not identified as being significantly regulated in the unbiased analysis of the RNA seq data. The focus analysis, however, provides some evidence that collagen remodeling is affected in lung aging, which is also confirmed by staining analysis of collagen deposition.

Critic: Overall this study provides a phenotypic analysis of lung aging in human tissue biopsy with a focus on known aging markers. I have some concerns that the analysis is not revealing new aspect, which may be due to the fact that the authors focus on the known biomarkers of lung aging. Also, the study completely lacks mechanistic studies to analyze the functional importance of the identified aging markers. As such, I am afraid that this purely descriptive study on human lung aging is not adding sufficient mechanistic new insights to our understanding of lung aging. Also, the study fails to cite some of the previous studies including data on human lung aging and COPD showing increases in senescent cells with increased expression of Cdkn2a (p16) and telomere-associated foci of DNA damage (TAFs) in human lung aging (see for example Birch et al. American Physiological Society 2015). I also have major concerns about the quality of the lung biopsies and that necropsy time may have an influence on the reported results.

Specific points:

Quality of the lung biopsies: This reviewer cannot clearly judge the quality of the lung samples that went into this study. The authors keep the description of the biopsies rather short in the method section: "RNA-seq, type 2 cell immunofluorescence, and telomere length analyses were done with the Lung Aging Cohort, which consists of 86 donor lungs collected between 2012 and 2018 and made available by the Donor West Network (37). Fresh tissue fragments were snap-frozen in liquid nitrogen within 48 hours of x-clamp." The authors side a paper from 2002, which of course cannot specify the collection of samples in the years 2012-2018. To me it is unclear where the biopsies derived from. From the name of the consortium, it can be assumed that the donor were people that died in accidents and where enrolled in organ donor programs. Is that assumption correct? It is also

not clear what the authors mean with “x-clamp”. Is that terminus referring to a surgical procedure to explant the donor lung? Samples were then taken with 48 hours after this procedure? How long was the timing to the start the explant-procedure after death of the donor had occurred? Why were biopsies taken from these explant lungs? What part of the lung was biopsied? Always the same region? To me it looks like a long time-window. 48 h can have a huge impact on RNA expression. What was the time-span of organ taking? The authors should conduct a correlation analysis on the observed regulation of RNA markers of lung aging vs. time-span of organ taking and sample freezing. It seems possible that some of the stress markers (senescence) may be induced by this time-span of necropsy.

Figure 2e: The authors describe in the result section “To test the significance of telomere shortening to cellular states, gene expression was compared between subsamples that were significantly different in telomere length but approximately matched by age.” The authors conduct a subsample analysis on 3 biopsies with very short and on 3 biopsies with very long telomeres in the age group of donors being 55-70 years old. The samples with short telomeres compared to the samples with long telomeres show more CDKN2A-positive cells in the representative immunohistochemistry stain, more gamma-H2AX foci (graph on the left), and in the IPA-analysis of the RNA-seq data an increase in p53 and Cdkn2a signaling. Following my argumentation above, it is well possible that the time-period to take and freeze organ biopsies (after death of the donor) was longer for those samples with shorter telomeres compared to those sample with longer telomeres. Therefore, the results may not be age-related but due to necropsy time. In addition, the analysis on the right (gammaH2aX) shows the result on individual cells, but as the authors describe they only analyzed 3 samples. It would be more appropriate to depict mean values per individua sample and to calculate the significance on the n=3 data points per group.

Figure 3a: The authors describe “Interestingly, the proportion of epithelial cells declined with age; on the other hand, the proportion of fibroblasts increased, consistent with fibrotic change in the aging lung.” I find this statement misleading. The analysis of the own data set is actually not supporting this conclusion. The r-square values are extremely low and p-values are not significant. The publicly available data seem to support the conclusion though. This has to be re-written. Also, I think that the authors could make more use of the single cell RNA analysis. I find this part of the data under-explored and not well presented in the current manuscript.

The focus on collagen deposition is solely based on the literature that shows accumulation of collagen deposition in human lung aging. The authors state: “We noted no robust differences in expression of collagen genes between old and young. However, collagen accumulation is due not simply to excess collagen deposition, but also to an imbalance of collagen production and destruction, as well as changes in extracellular collagen structure and stability^{29,30}. Therefore, the oldest and youngest quintiles in the LAC were examined for expression of genes known to regulate post-translational processing of collagen, including lysyl oxidases, transglutaminases, and tissue inhibitors of matrix metalloproteases, which inhibit collagen turnover by metalloproteases.” While I

find the results interesting and supported by the existing literature, I think that it would be better the authors would focus on an unbiased analysis of the data sets. I think this may reveal more interesting new aspects of lung aging. However, the authors need to control the quality of the biopsies as the time of necropsy may have a significant influence on the results. Along these lines the IPA analysis of RNA seq data revealed “Necrosis” and “Apoptosis” as the strongest enriched GO-terms associated with aging (Fig. 2C). These processes are known to be induced by tissue necrosis after death. It is well possible that aging increases the vulnerability of tissues to increases in necropsy time. This could be tested experimentally in animal models.

Reviewer #2 (Remarks to the Author):

In this study, Lee and colleagues molecularly characterize age-related alterations in human lungs. The authors prospectively collected 86 human donor lungs for the donors between 16 and 76 years, and performed immunostaining, RNAseq, telomere length-specific PCR and other assays to study their samples. Where possible, an attempt was made to compare the results obtained with the literature GTEx dataset. I find this manuscript valuable as it provides additional information about aging lung at the molecular level. However, in the present form the manuscript lacks sufficient parts of data analysis and must be improved in several parts.

- 1) Abstract is not sufficiently informative, and I recommend to concentrate more on the original study results characterized by the numbers and avoid general statements
- 2) The authors should adequately reference previous studies on molecular feature/pathway analyses linked with lung aging and fibrosis, and compare the results obtained with the previous findings. This is not at all the case in the current version.
- 3) Molecular pathway analysis by IPA is scarce and not sufficient. It could be expanded by using RNAseq data to calculate direct pathway activation levels for aged vs not aged lungs in few thousand molecular pathways to interrogate bigger repertoire of signaling, metabolic, and other pathways that could be more insightful. For example, that could be implemented for ~3,000 human pathways using the recently published script and database (<https://doi.org/10.3389/fgene.2021.617059>).
- 4) I also suggest performing gene ontology terms analysis to expand the functional assay and to cross-link the results with the pathway analysis, see above
- 5) The discussion lacks practical implications of the data presented. It should be expanded to add Connectivity Map tool analyses for gene expression data that might suggest possible strategies of molecular intervention to improve aged lung conditions.

6) Please update figures according to the results obtained with reference to the previous points 3-5

7) Direct item-by-item comparisons with the GTEx tissue datasets should be presented as the supplementary materials

8) RNA sequencing and data quality. What kits were used for mRNA enrichment and cDNA preparation? How many gene-mapped sequencing reads were obtained per library? Were there any replicates to assess reproducibility and data consistency?

9) Ethical statement. I am not certain that "Tissue samples were obtained from brain-dead (deceased) individuals, and thus this study does not qualify as human subjects research, as confirmed by the UCSF and Columbia University IRBs" is enough. Could the authors please expand this formulation.

10) Sequencing data availability. These data must become freely publicly available following NPG policy; however, I was not able to access the RNAseq data for this manuscript. Could the authors please provide detailed instructions. Do they plan to make the data publicly available after publication? In principle, this should be a prerequisite of publication.

Reviewer #3 (Remarks to the Author):

This is an interesting study that utilized the Lung Aging Cohort and publicly available data (GTEx) to characterize normal lung aging. Signatures of cellular senescence (in particular, p16 and p53 driven pathways) and fibrosis were shown to characterize the aging process in lungs. Although the studies are well done, and data are interesting, some concerns need to be considered.

1. The studies involving second harmonic (SH) generation imaging (Fig. 4) are difficult to interpret. The number of samples in the aged (51-86 years) group is low (n=4), and within each subject, there is likely to be significant variation between upper and lower lobes, and between conducting airways gas-exchange regions. This must be more carefully considered since there is data supporting the notion that "normal aging" results in an emphysema phenotype, rather than fibrosis. How does one reconcile that telomerase deficiency syndromes can lead to emphysema in some and fibrosis in others.

2. Similarly, it is interesting that lung aging correlates more with skin aging, at a molecular level, despite the finding that most cases of human skin aging results in loss of ECM, reduced elasticity, and "wrinkling". Is there an increase in fibroblasts and loss of keratinocytes in aging skin, as found for fibroblast-epithelial imbalance in aging lung?

We thank the reviewers for their comments, which have enhanced the manuscript in revision. Our responses are indicated in bold below.

-Mallar Bhattacharya, on behalf of all the authors

Reviewer #1 (Remarks to the Author):

I have some concerns that the analysis is not revealing new aspect, which may be due to the fact that the authors focus on the known biomarkers of lung aging. Also, the study completely lacks mechanistic studies to analyze the functional importance of the identified aging markers. As such, I am afraid that this purely descriptive study on human lung aging is not adding sufficient mechanistic new insights to our understanding of lung aging...The focus on collagen deposition is solely based on the literature that shows accumulation of collagen deposition in human lung aging. The authors state: "We noted no robust differences in expression of collagen genes between old and young. However, collagen accumulation is due not simply to excess collagen deposition, but also to an imbalance of collagen production and destruction, as well as changes in extracellular collagen structure and stability^{29,30}. Therefore, the oldest and youngest quintiles in the LAC were examined for expression of genes known to regulate post-translational processing of collagen, including lysyl oxidases, transglutaminases, and tissue inhibitors of matrix metalloproteases, which inhibit collagen turnover by metalloproteases." While I find the results interesting and supported by the existing literature, I think that it would be better the authors would focus on an unbiased analysis of the data sets. I think this may reveal more interesting new aspects of lung aging.

We thank the reviewer for raising the question of how our study adds to the understanding of lung aging and feel that there are several novel aspects that advance the field. Much of the lung aging data in the literature have been developed in nonhuman models. To our knowledge, the manuscript is the first to analyze the effects of age on the human lung with transcriptomics, telomere length measurement, and functional imaging with multiple datasets (the UCSF lung aging cohort, the Columbia cohort, and GTEx lung), in a large number of samples. We are aware of one recent study that has analyzed the effects of human lung aging on gene expression (Chow et al PMID: 3339797), but this study was focused on SARS-CoV2 pathogenesis, lacked validation in multiple datasets as in our study, and analyzed gene expression alone, whereas our study takes the analysis into new terrain with multimodal approaches as detailed below and highlighted in the discussion of the manuscript:

- **First, our analysis defines the most differentially expressed genes with lung aging, which have not previously been reported and will serve as novel lung aging biomarkers.**
- **Second, we address the question of whether lung aging is associated with cellular senescence, which has been studied most extensively in cultured cells and animal models rather than human tissues. As the reviewer points out, we did confirm known markers such as P16 and P21. However, we also applied a novel 11-gene cellular senescence score that we recently published (DePianto et al., JCI Insight PMID: 33705361). These genes were acquired by analysis of multiple cultured epithelial and mesenchymal cell lines, most of which were lung-derived and not previously known. An unmet need in the field of cellular senescence is to confirm the relevance of cellular senescence to human tissues, and our manuscript provides the first evidence in a large sample size of increasing expression with age of both canonical and novel markers of senescence in the human lung.**
- **Third, with reference to a potential cause of the senescence profile observed, we present evidence for telomere shortening in the lung in a range that we found was associated with activation of upstream regulators of senescence. Thus, the results provide evidence for the role of telomere shortening in driving the senescence profile**

in the lung, an analysis again that is without precedent to our knowledge for normal aging.

- Finally, regarding the focus on collagen regulatory genes, it was in fact our *unbiased* analysis of the RNAseq data that led to the discovery of increased pro-fibrotic pathways in the lung with age, including TGFbeta-related signaling mediators and increased fibroblasts; this finding in turn motivated the hypothesis that collagen-regulatory genes may be increased, given their importance in the literature for pathologic lung fibrosis. A study of their expression in normal lung aging has not previously been undertaken. The upregulation of collagen-regulatory genes is taken together with aging-associated fibrotic changes by second harmonic imaging, which we have shown for the first time limited alveolar expansion and surfactant secretion, providing a multimodal and functional perspective on the effects of the fibrotic program in human lung aging. These findings with normal lung aging have particular significance in light of the recent emergence of subclinical, asymptomatic fibrosis by imaging in aged individuals (interstitial lung abnormalities), for which we provide a potential molecular correlate. While testing the effects of individual mediators is beyond the scope of the current study, a point that we have presented in the revision as a limitation and an opportunity for future study, the aging programs in the lung revealed by our manuscript will fill a gap in knowledge about the genetic and, from our imaging results, functional consequences of lung aging.

Also, the study fails to cite some of the previous studies including data on human lung aging and COPD showing increases in senescent cells with increased expression of Cdkn2a (p16) and telomere-associated foci of DNA damage (TAFs) in human lung aging (see for example Birch et al. American Physiological Society 2015).

Regarding the Birch et al. study, that study only reports telomere length in diseased individuals (COPD patients). Therefore, we did not cite this study because we are focused on aging in the healthy lung. We also respectfully point out that our study, to our knowledge, represents the first large-scale study of human lung aging that includes both gene expression data, telomere length analysis, and parenchymal imaging. Still, to contextualize the work in relation to the literature, we have added references to relevant previous studies in the introductory paragraphs to provide context (lines 48-52).

I also have major concerns about the quality of the lung biopsies and that necropsy time may have an influence on the reported results. Quality of the lung biopsies: This reviewer cannot clearly judge the quality of the lung samples that went into this study. The authors keep the description of the biopsies rather short in the method section: "RNA-seq, type 2 cell immunofluorescence, and telomere length analyses were done with the Lung Aging Cohort, which consists of 86 donor lungs collected between 2012 and 2018 and made available by the Donor West Network (37). Fresh tissue fragments were snap-frozen in liquid nitrogen within 48 hours of x-clamp." The authors cite a paper from 2002, which of course cannot specify the collection of samples in the years 2012-2018. To me it is unclear where the biopsies derived from. From the name of the consortium, it can be assumed that the donor were people that died in accidents and where enrolled in organ donor programs. Is that assumption correct? It is also not clear what the authors mean with "x-clamp". Is that terminus referring to a surgical procedure to explant the donor lung? Samples were then taken with 48 hours after this procedure? How long was the timing to the start the explant-procedure after death of the donor had occurred? Why were biopsies taken from these explant lungs? What part of the lung was biopsied? Always the same region? To me it looks like a long time-window. 48 h can have a huge impact on RNA expression. What was the time-span of organ taking? The authors should conduct a correlation analysis on the observed regulation of RNA markers of lung aging vs.

time-span of organ taking and sample freezing. It seems possible that some of the stress markers (senescence) may be induced by this time-span of necropsy...the authors need to control the quality of the biopsies as the time of necropsy may have a significant influence on the results. Along these lines the IPA analysis of RNA seq data revealed "Necrosis" and "Apoptosis" as the strongest enriched GO-terms associated with aging (Fig. 2C). These processes are known to be induced by tissue necrosis after death. It is well possible that aging increases the vulnerability of tissues to increases in necropsy time. This could be tested experimentally in animal models.

We appreciate the reviewer's comment and are glad to address this issue in the revision. These donors are either brain-dead or donors after cardiac death. Cross-clamp (x-clamp) time refers to clamping of the aorta in the operating room for cessation of blood flow and tissue perfusion, which enables tissue harvest. We agree with the reviewer's excellent suggestion to rule out the possibility of an effect of duration from tissue harvest to cryopreservation on gene expression.

-First, we have added a reference in the text to our previous study of 99 lungs in which we found that cold ischemia time did not have any effect on the level of cytokines produced by the intact lung in response to pro-inflammatory stimuli delivered by whole-lung perfusion (Leligdowicz et al, PMID: 32955627), in the range of duration that was similar to our present study (20 +/- SD 13.5 h).

-Second, we tested whether time from cross-clamp to cryopreservation may unexpectedly have been longer for the aged samples by chance; while these data were not available for all the samples in our cohort, for those samples for which the data were available, we plotted age against time to cryopreservation and found no significant correlation, as shown to the right. These data have been added to Supplementary Figure S1e.

-Third, we considered the possibility that senescence markers may relate to time to cryopreservation. We therefore plotted CDKN2A and CDKN1A against time to cryopreservation and found no significant correlation for either, suggesting that these markers are not increased by longer time to cryopreservation. These data, shown to the right, have also been added to Supplementary Figure S1e.

-Fourth, regarding time to cryopreservation and cell death, we likewise tested whether the expression of genes listed by IPA under the apoptosis and necrosis pathways were correlated with age in the samples for which we have times to cryopreservation. Our analysis shown below revealed no significant increase or decrease in these genes, computed as a unified gene score, with increasing time to cryopreservation, indicating that within the range of time to cryopreservation in our study, there was no effect on

genes relating to cell death. These data indicate that within the span of the range of time to cryopreservation in our study and given maintenance of the lungs on ice during this period, no induction of cell death pathways is induced by cryopreservation itself. The plots have been added to Supplementary Figure S1e.

Figure 2e: The authors describe in the result section “To test the significance of telomere shortening to cellular states, gene expression was compared between subsamples that were significantly different in telomere length but approximately matched by age.” The authors conduct a subsample analysis on 3 biopsies with very short and on 3 biopsies with very long telomeres in the age group of donors being 55-70 years old. The samples with short telomeres compared to the samples with long telomeres show more CDKN2A-positive cells in the representative immunohistochemistry stain, more gamma-H2AX foci (graph on the left), and in the IPA-analysis of the RNA-seq data an increase in p53 and Cdkn2a signaling. Following my argumentation above, it is well possible that the time-period to take and freeze organ biopsies (after death of the donor) was longer for those samples with shorter telomeres compared to those sample with longer telomeres. Therefore, the results may not be age-related but due to necropsy time. In addition, the analysis on the right (gammaH2ax) shows the result on individual cells, but as the authors describe they only analyzed 3 samples. It would be more appropriate to depict mean values per individual samples and to calculate the significance on the n=3 data points per group.

We agree that time to cryopreservation could also affect DNA damage responses. Time to cryopreservation data could not be compared for the samples corresponding the short versus long telomere analysis because these times were not available for all of the samples. Therefore, instead, we tested the effect of time to cryopreservation on DNA damage foci in samples for which these times were available. The plot to the left shows results for gamma H2ax labeling of DNA damage foci for individual samples chosen widely divergent length of time to cryopreservation but matched for age, sex, and telomere length. 6, 7, and 9-hr samples are plotted in the “short” category, and 32, 36, and 42-h samples are grouped in the “long”

category. There was no effect of time to cryopreservation on the number of gamma H2ax foci per HPF. This plot has been added to Supplementary Figure S1d.

Regarding statistical testing for differences between groups, comparing means alone does not capture the variance of the individual measurements for each lung donor, which represent important features of biological as opposed to technical variance. To account for the unequal variances between the two groups, in our revision we have conducted the Welch's t-test for significance testing as shown to the right.

Figure 3a: The authors describe “Interestingly, the proportion of epithelial cells declined with age; on the other hand, the proportion of fibroblasts increased, consistent with fibrotic change in the aging lung.” I find this statement misleading. The analysis of the own data set is actually not supporting this conclusion. The *r*-square values are extremely low and *p*-values are not significant. The publicly available data seem to support the conclusion though. This has to be re-written. Also, I think that the authors could make more use of the single cell RNA analysis. I find this part of the data under-explored and not well presented in the current manuscript.

We thank the reviewer for this comment. As the reviewer mentions, the findings for our cohort were validated by analysis of the publicly available GTEx cohort. The GTEx dataset has a larger sample size (N=345) than our own and is thus better powered to detect differences, which were at a trend level in our cohort. However, we would like to point out that immunofluorescence staining of samples confirmed the finding of decreasing epithelial cells. Furthermore, thanks to the reviewer’s comment, we advanced our analysis to address the question of which subtypes of epithelial cell proportions change over time using another single cell RNA-seq dataset that has annotations of lung cellular subtypes (Travaglini et al, PMID: 33208946). Consistent with our previous analysis, the proportion of alveolar fibroblasts increased and the proportion of alveolar type 2 cells, an epithelial subtype, decreased overtime (these data have been added to the supplement, specifically Figure S2c). Similar to our previous analysis, the epithelial subtype results were not significant in our own dataset but displayed a concordant trend with the larger GTEx dataset. Given the concordance and consistency of the data across the two cohorts and across analyses, we have retained the analyses of both our cohort and GTEx.

Reviewer #2 (Remarks to the Author):

1) Abstract is not sufficiently informative, and I recommend to concentrate more on the original study results characterized by the numbers and avoid general statements

We thank the reviewer for this suggestion and have re-written the abstract with more specific quantitative detail pertaining to our main findings.

2) The authors should adequately reference previous studies on molecular feature/pathway analyses linked with lung aging and fibrosis, and compare the results obtained with the previous findings. This is not at all the case in the current version.

We thank the reviewer for this suggestion, and as mentioned above in our comments for Reviewer #1, the number of studies of human lung aging are few and limited in scope; but to contextualize the work in relation to the literature, we have added references to relevant previous studies (mostly mouse) in the introductory paragraphs (lines 48-52).

We agree that further reference to relevant aspects of the pathologic lung fibrosis literature would enhance our discussion. In addition to reference to the literature on telomere shortening and lung fibrosis (line 195), we have added discussion and references to collagen regulatory genes that we found to be upregulated with age and that have been tested in fibrosis models and found to have pro-fibrotic effects (lines 207-209).

3) Molecular pathway analysis by IPA is scarce and not sufficient. It could be expanded by using RNAseq data to calculate direct pathway activation levels for aged vs not aged lungs in few thousand molecular pathways to interrogate bigger repertoire of signaling, metabolic, and other pathways that could be more insightful. For example, that could be implemented for ~3,000 human pathways using the recently published script and database.

<https://doi.org/10.3389/fgene.2021.617059>. I also suggest performing gene ontology terms analysis to expand the functional assay and to cross-link the results with the pathway analysis, see above.

We thank the reviewer for this suggestion of including other bioinformatic approaches to confirm the results of IPA. We performed Gene Ontology enrichment analysis with the top positive and negative age-correlated genes. The upregulated genes analysis revealed pathways related to cell communication/movement, signal transduction, and response to stimulus. The downregulated genes analysis confirmed our findings from IPA regarding decreased cell cycle and DNA repair pathways with age. These analyses, shown below, were added to Supplementary Figure 1c and Supplementary Table 5.

5) The discussion lacks practical implications of the data presented. It should be expanded to add Connectivity Map tool analyses for gene expression data that might suggest possible strategies of molecular intervention to improve aged lung conditions.

We thank the reviewer for this interesting observation. Aging has recently been coming under attention as a potential therapeutic target, although conventions around how to therapeutically target normal, age-related, progressive loss of physiologic function of tissues, such as is known to occur in the lung, remain unspecified. For example, aging per se and associated conditions are not in most cases identifiable among indications recognized by national drug approval agencies. However, we agree that in future this could be an exciting application of the results we have defined for lung aging and senescence, which corresponded to physiologic dysfunction in our imaging studies. Nonetheless, as suggested by the reviewer, the Aging and Collagen signature gene sets from our study were interrogated for potential therapeutic drug candidates. These data, shown below, have been added to Supplementary Figure S3c and full results are listed in Supplementary Table 6.

6) Please update figures according to the results obtained with reference to the previous points 3-5.

The figures have been updated to reflect the revisions pertaining to comments 3-5.

7) Direct item-by-item comparisons with the GTEx tissue datasets should be presented as the supplementary materials

We have moved the item-by-item comparisons to the supplementary figures—specifically, Figure S1a and S1b.

8) *RNA sequencing and data quality. What kits were used for mRNA enrichment and cDNA preparation? How many gene-mapped sequencing reads were obtained per library? Were there any replicates to assess reproducibility and data consistency?*

mRNA was enriched using oligo(dT) beads using the NEBNext Poly(A) mRNA Magnetic Isolation Module (New England Biolabs), followed by random fragmentation. cDNA was synthesized using the NEBNext Ultra II RNA Library Prep Kit for Illumina (New England Biolabs). These details have been added to the Methods section of the manuscript. The number of uniquely mapped reads ranged from 19.1 million to 42.7 million with an average of 25.3 million reads per sample. We have added this information to the manuscript. There were no technical replicates but we do note that we validated findings in a separate cohort (GTex).

9) *Ethical statement. I am not certain that "Tissue samples were obtained from brain-dead (deceased) individuals, and thus this study does not qualify as human subjects research, as confirmed by the UCSF and Columbia University IRBs" is enough. Could the authors please expand this formulation.*

The samples in the study including both UCSF and Columbia were acquired from deceased individuals, and thus the study is not considered human subjects research as per United States Department of Health and Human Services human subject regulations under 45 CFR 46. We have clarified this information in the text in the Methods section.

10) *Sequencing data availability. These data must become freely publicly available following NPG policy; however, I was not able to access the RNAseq data for this manuscript. Could the authors please provide detailed instruction. Do they plan to make the data publicly available after publication? In principle, this should be a prerequisite of publication.*

We have deposited the data at NCBI's sequence read archive under the dbGAP, and the data will be made publicly available at publication.

Reviewer #3 (Remarks to the Author):

1. *The studies involving second harmonic (SH) generation imaging (Fig. 4) are difficult to interpret. The number of samples in the aged (51-86 years) group is low (n=4), and within each subject, there is likely to be significant variation between upper and lower lobes, and between conducting airways gas-exchange regions. This must be more carefully considered since there is data supporting the notion that "normal aging" results in an emphysema phenotype, rather than fibrosis. How does one reconcile that telomerase deficiency syndromes can lead to emphysema in some and fibrosis in others.*

We thank the reviewer for several insightful comments and have responded below to each in sequence.

"The number of samples in the aged group (51-86 years) is low": We increased the lung numbers in the aged group, as suggested (revised Figure 4). The difference between groups is still significant, and our original conclusions stand.

"there is likely to be significant variation between upper and lower lobes, and between conducting airways gas-exchange regions": This is an interesting question that we have not been able to address currently for technical reasons and should be addressed by future studies. A main goal of the SHG studies was to establish a unique structure-function relationship between peri-alveolar collagen and type 2 cell function, assessed as surfactant secretion, that might apply to all lung lobes irrespective of collagen variations. The linear structure-function relationship we report reveals a previously unknown

inhibitory effect of peri-alveolar collagen accumulation on surfactant secretion. We were able to establish this understanding, because we directly viewed the live lung by optical microscopy and could therefore, quantify type 2 cell functionality across a wide range of collagen expressions in different alveoli. The downside of live optical imaging was that we were restricted by the stage dimensions of the microscope to viewing the relatively thin lingular lobe, and could not accommodate the bulkier upper and lower lobes under the objective. Despite these constraints, we believe our findings signify the need to interpret structural changes in the context of cell function to better understand the lung's aging responses. We point out that our SHG findings are alveolus specific, and do not apply to airway collagen, which is unlikely to be a determinant of type 2 cell secretion.

“there is data supporting the notion that “normal aging” results in an emphysema phenotype, rather than fibrosis. How does one reconcile that telomerase deficiency syndromes can lead to emphysema in some and fibrosis in others”: We thank the reviewer for this interesting perspective, though we did not detect an emphysema phenotype in the imaging data. However, we agree that it is possible a global analysis of the entire lung by other methods, such as CT scanning, would reveal aging-associated emphysema in addition to the collagen we have detected at the local scale. If such emphysematous regions were identified, in future studies could apply our methods to assess the effect on surfactant secretion of type 2 cell function. With regard to the telomerase deficiency syndromes, since our study is of normal aging, we cannot comment on that pathologically specific circumstance, which is beyond the scope of our sample. The revised text addresses these issues (lines 210-226).

2. Similarly, it is interesting that lung aging correlates more with skin aging, at a molecular level, despite the finding that most cases of human skin aging results in loss of ECM, reduced elasticity, and “wrinkling”. Is there is an increase in fibroblasts and loss of keratinocytes in aging skin, as found for fibroblast-epithelial imbalance in aging lung?

We thank the reviewer for this interesting suggestion and performed a cell type deconvolution analysis with skin cell types defined with SingleR for scRNAseq data from a recent single cell paper (He et al, PMID: 32035984). This analysis did indeed reveal an increase in fibroblasts for sun exposed but not non-sun exposed skin. While as the reviewer points out the skin has other specific features such as laxity that may not be relevant to the lung biology featured in our paper, the results are of interest to a general audience and have been added to Supplementary Figure S2c.

REVIEWER COMMENTS

Reviewer #1 (Remarks to the Author):

Mallar Bhattacharya et al. Have addressed some of my concerns and the study has improved. I have some remaining questions of points that I think are still not adequately addressed. Moreover, I find the mechanistic data and single cell data still weak and study remains descriptive. I agree that it is interesting to the field of lung research as it is a comprehensive description on age related changes.

The authors argument on previous studies and COPD: I don't think the line of argumentation is sound. COPD is regarded as an aging associated lung disease and as such it is an important condition to be investigated in studies on lung aging. The authors should discuss the study of Birch et al. in greater detail. It would also be interesting to discuss the study of Tsui et al. 2006 (Alveolar cell senescence in patients with pulmonary emphysema).

Tsuji T, Aoshiha K, Nagai A. Am J Respir Crit Care Med. 2006 Oct 15;174(8):886-93. doi: 10.1164/rccm.200509-1374OC. Epub 2006 Aug 3.) The authors investigated COPD compared to non-symptomatic non-smokers and smokers. The study also shows an increase in senescence in aging, especially in COPD. I find this an important point since it indicates that aging of the lung may lead to COPD, which as such could be seen as an aging associated disease. I don't understand why it is important to show in a very small sample cohort that aging markers (senescence) also associate with lung aging in the absence of COPD in this small group on investigated people. The authors should make this more clear.

Regarding the quality of sample taking: I think that the new data are very informative and illustrate that time to cryo-conversation had no major influence on a set of target genes that the authors found to be dysregulated during aging, including senescence associated genes. This is important new data that make the study better.

Figure 2e: I still think that the authors should present an extra panel on actual sample number (3 dots per group) plus minus SD, plus p-value. Just by looking at many cells per sample and to take each cell as a biological replicate, it is very easy to get significant differences. However, whether these differences are useful to discriminate individuals with lung aging from those with less lung aging has to be addressed in a different way as I suggested.

Influence of cryopreservation on DNA damage foci: This new data I do not understand. The authors describe in the rebuttal-figure (new Suppl. Fig. S1d): "The plot to the left shows results for gamma

H2ax labeling of DNA damage foci for individual samples chosen widely divergent length of time to cryopreservation but matched for age, sex, and telomere length. 6, 7, and 9-hr samples are plotted in the “short” category, and 32, 36, and 42-h samples are grouped in the “long” category.”

I don't understand this. Why does the “long category” also includes dots of 6h sample (dark blue)? If those 2 sample would be removed, wouldn't the “long category” show less gH2AX foci? What does that mean?

Reviewer #2 (Remarks to the Author):

Thank you for reacting on most of my comments. However, one item from previous #3 was not addressed and needs to be improved:

"3) Molecular pathway analysis by IPA is scarce and not sufficient. It could be expanded by using RNAseq data to calculate direct pathway activation levels for aged vs not aged lungs in few thousand molecular pathways to interrogate bigger repertoire of signaling, metabolic, and other pathways that could be more insightful. For example, that could be implemented for ~3,000 human pathways using the recently published script and database.

(<https://doi.org/10.3389/fgene.2021.617059>). I also suggest performing gene ontology terms analysis to expand the functional assay and to cross-link the results with the pathway analysis, see above.

We thank the reviewer for this suggestion of including other bioinformatic approaches to confirm the results of IPA. We performed Gene Ontology enrichment analysis with the top positive and negative age-correlated genes. The upregulated genes analysis revealed pathways related to cell communication/movement, signal transduction, and response to stimulus. The downregulated genes analysis confirmed our findings from IPA regarding decreased cell cycle and DNA repair pathways with age. These analyses, shown below, were added to Supplementary Figure 1c and Supplementary Table 5."

-I still believe that the pathway analysis in the present form is not strong enough, and more pathways (including metabolic and repair pathways), and quantitative pathway activation metrics need to be investigated/identified/discussed, as suggested in my previous report.

Reviewer #3 (Remarks to the Author):

The authors have responded satisfactorily to my concerns.

We thank the reviewers for their comments on our first revision and appreciate their further advice, which we have incorporated into the current revision. The reviewer comments are copied below with our responses indicated in bold, and we have highlighted changes in the manuscript.

-Mallar Bhattacharya, on behalf of all the authors

Reviewer #1 (Remarks to the Author):

Mallar Bhattacharya et al. Have addressed some of my concerns and the study has improved. I have some remaining questions of points that I think are still not adequately addressed. Moreover, I find the mechanistic data and single cell data still weak and study remains descriptive. I agree that it is interesting to the field of lung research as it is a comprehensive description on age related changes. The authors argument on previous studies and COPD: I don't think the line of argumentation is sound. COPD is regarded as an aging associated lung disease and as such it is an important condition to be investigated in studies on lung aging. The authors should discuss the study of Birch et al. in greater detail. It would also be interesting to discuss the study of Tsui et al. 2006 (Alveolar cell senescence in patients with pulmonary emphysema. Tsuji T, Aoshiba K, Nagai A. Am J Respir Crit Care Med. 2006 Oct 15;174(8):886-93. doi: 10.1164/rccm.200509-1374OC. Epub 2006 Aug 3.) The authors investigated COPD compared to non-symptomatic non-smokers and smokers. The study also shows an increase in senescence in aging, especially in COPD. I find this an important point since it indicates that aging of the lung may lead to COPD, which as such could be seen as an aging associated disease. I don't understand why it is important to show in a very small sample cohort that aging markers (senescence) also associate with lung aging in the absence of COPD in this small group on investigated people. The authors should make this more clear.

Thank you for underscoring the importance of COPD, another major lung disease besides lung fibrosis where senescence is of importance. While our findings were more relevant to fibrosis, we agree that discussing COPD could be helpful for some readers as context for our study and have added the following text to the discussion (directly following mention of the relevance of our study to lung fibrosis): “We also note that senescence has been recognized as an important feature of chronic obstructive pulmonary disease (COPD), where telomere dysfunction or shortening has been associated with DNA damage foci and expression of senescence markers⁴⁴[Birch et al],⁴⁵ [Tsuji et al]. Under which conditions senescence in the aging lung may contribute to the severity of lung fibrosis or COPD, or both, could be addressed by prospective studies comparing healthy, fibrotic, and COPD cohorts directly.” (Lines 209-214)

With regard to the reviewer's broader concern about the importance or impact of the study of normal lung aging, we respectfully point out that data about the molecular natural history of the lung are lacking; our study therefore fills this knowledge gap. We also note that, including our validation analysis with GTEx and our separate cohort for imaging studies, we make use of analyses from over 400 individuals. But we appreciate the reviewer's suggestion to clarify the importance of lung aging. Our perspective is that the literature on senescence in lung disease, including the cited paper by Tsujino et al, suggests that cellular senescence may drive lung dysfunction in health as well as disease progression. Our study, therefore, provides an important point of reference for understanding senescence from the perspective of disease risk; that is, our finding of

senescence by RNAseq and physiologic dysfunction by functional microscopy reveals senescence as a vulnerability factor present even in the healthy lung as it ages. We have edited our discussion to make these points more clearly: “Our study sheds light on lung aging, making use of transcriptomic as well functional analyses from three cohorts spanning the adult age range including over 400 individuals. The perspective gained by our findings is that cellular senescence may drive lung dysfunction. That is, our finding of senescence and fibrosis signatures by RNAseq and physiologic dysfunction associated with increased collagen density by microscopy reveals senescence as a potential vulnerability factor present even in the healthy lung as it ages. Future studies might build on these findings by exploring the functional effects of the identified biomarkers and mediators of lung aging, determining the relative weight of cell autonomous and environmental effects, and testing whether pre-existing cellular senescence is a predictive factor for the development of, or contributes to worse outcomes in, acute and chronic lung diseases.” (Lines 241-250)

Regarding the quality of sample taking: I think that the new data are very informative and illustrate that time to cryo-conversation had no major influence on a set of target genes that the authors found to be dysregulated during aging, including senescence associated genes. This is important new data that make the study better.

We are glad that these analyses have been helpful.

Figure 2e: I still think that the authors should present an extra panel on actual sample number (3 dots per group) plus minus SD, plus p-value. Just by looking at many cells per sample and to take each cell as a biological replicate, it is very easy to get significant differences. However, whether these differences are useful to discriminate individuals with lung aging from those with less lung aging has to be addressed in a different way as I suggested.

The reviewer raises an interesting question that relates to our core perspective on telomere biology. We feel that the relevant comparison for this analysis is across all anatomic samplings (ie, all high-power fields). The reason is that, unlike height or weight, which are unitary properties of individuals at the whole organism level, DNA damage in senescence as measured by gamma H2ax varies across single cells within individuals. Therefore, appreciation of the relationship between telomere length and the DNA damage response is better viewed and quantified at the cellular level. Nevertheless, we think it is reasonable to present the aggregate individual level data as well, and this analysis, shown above, was added as Figure S1f: “IPA upstream regulator analysis of differentially expressed genes revealed that the canonical senescence regulators p53 and p16 were activated in association with decreased telomere length; furthermore, sites of DNA damage were significantly increased across multiple high power fields for the short-compared to long-telomere samples by gamma-H2ax immunohistochemistry (Figure 2d), with a similar trend when the data for individuals were averaged (Figure S1f).”

Influence of cryopreservation on DNA damage foci: This new data I do not understand. The authors describe in the rebuttal-figure (new Suppl. Fig. S1d): "The plot to the left shows results for gamma H2ax labeling of DNA damage foci for individual samples chosen widely divergent length of time to cryopreservation but matched for age, sex, and telomere length. 6, 7, and 9-hr samples are plotted in the "short" category, and 32, 36, and 42-h samples are grouped in the "long" category." I don't understand this. Why does the "long category" also includes dots of 6h sample (dark blue)? If those 2 sample would be removed, wouldn't the "long category" show less gH2AX foci? What does that mean?

Thank you so much for pointing this out: the dark blue samples were erroneously added by the software program and are not part of the data. We have corrected the error in the updated Figure S1d (now Figure S1e) and apologize for the confusion.

Reviewer #2 (Remarks to the Author):

Thank you for reacting on most of my comments. However, one item from previous #3 was not addressed and needs to be improved:

"3) Molecular pathway analysis by IPA is scarce and not sufficient. It could be expanded by using RNAseq data to calculate direct pathway activation levels for aged vs not aged lungs in few thousand molecular pathways to interrogate bigger repertoire of signaling, metabolic, and other pathways that could be more insightful. For example, that could be implemented for ~3,000 human pathways using the recently published script and database. (<https://doi.org/10.3389/fgene.2021.617059>). I also suggest performing gene ontology terms analysis to expand the functional assay and to cross-link the results with the pathway analysis, see above. We thank the reviewer for this suggestion of including other bioinformatic approaches to confirm the results of IPA. We performed Gene Ontology enrichment analysis with the top positive and negative age-correlated genes. The upregulated genes analysis revealed pathways related to cell communication/movement, signal transduction, and response to stimulus. The downregulated genes analysis confirmed our findings from IPA regarding decreased cell cycle and DNA repair pathways with age. These analyses, shown below, were added to Supplementary Figure 1c and Supplementary Table 5." I still believe that the pathway analysis in the present form is not strong enough, and more pathways (including metabolic and repair pathways), and quantitative pathway activation metrics need to be investigated/identified/discussed, as suggested in my previous report.

We thank the reviewer for suggesting analysis by pathway activation level and have completed this analysis for the revision. The results were informative and support a pro-fibrotic transcriptomic profile of the aged lung, as described below in the revised manuscript: "We also performed pathway activation level analysis utilizing over 3,044 human molecular pathways extracted from the Biocarta, Reactome, KEGG, Qiagen Pathway Central, NCI, and HumanCYC databases³¹[Sorokin et al], comparing the oldest and youngest quintiles in the LAC and GTEx lung, and found that multiple growth factor pathways implicated in fibrosis, such as PDGF, FGF, LPA, and ephrin A³²⁻³⁵[Please see manuscript for refs], were increased in the aged lungs (Figure S1g and Table S4)." (Lines 107-112)

Reviewer #3 (Remarks to the Author):

The authors have responded satisfactorily to my concerns.

We thank the reviewer for their feedback on our manuscript.

REVIEWERS' COMMENTS

Reviewer #1 (Remarks to the Author):

No further comments

Reviewer #2 (Remarks to the Author):

Thank you, my comments were addressed now. However, a new minor error appeared: the pathways listed were most probably not "increased", but were upregulated, or activated

Reviewer #3 (Remarks to the Author):

The authors have satisfactorily addressed my concerns.

We thank the reviewers for their comments on our second revision. The reviewer comments are copied below with our response indicated in bold, and we have highlighted the change in the manuscript.

-Mallar Bhattacharya, on behalf of all the authors

Reviewer #1 (Remarks to the Author):

No further comments

Reviewer #2 (Remarks to the Author):

Thank you, my comments were addressed now. However, a new minor error appeared: the pathways listed were most probably not "increased", but were upregulated, or activated.

-We have changed the wording to "upregulated" in the text (line 119).

Reviewer #3 (Remarks to the Author):

The authors have satisfactorily addressed my concerns.